# Multispectral Food Classification and Caloric Estimation Using Convolutional Neural Networks

**DOI:** 10.3390/foods12173212

**Published:** 2023-08-25

**Authors:** Ki-Seung Lee

**Affiliations:** Department of Electrical and Electronic Engineering, Konkuk University, 1 Hwayang-dong, Gwangjin-gu, Seoul 05029, Republic of Korea; kseung@konkuk.ac.kr; Tel.: +82-2-450-3489

**Keywords:** multispectral imaging, convolutional neural network, food analysis, non-invasive analysis, dietary assessment, data fusion

## Abstract

Continuous monitoring and recording of the type and caloric content of ingested foods with a minimum of user intervention is very useful in preventing metabolic diseases and obesity. In this paper, automatic recognition of food type and caloric content was achieved via the use of multi-spectral images. A method of fusing the RGB image and the images captured at ultra violet, visible, and near-infrared regions at center wavelengths of 385, 405, 430, 470, 490, 510, 560, 590, 625, 645, 660, 810, 850, 870, 890, 910, 950, 970, and 1020 nm was adopted to improve the accuracy. A convolutional neural network (CNN) was adopted to classify food items and estimate the caloric amounts. The CNN was trained using 10,909 images acquired from 101 types. The objective functions including classification accuracy and mean absolute percentage error (MAPE) were investigated according to wavelength numbers. The optimal combinations of wavelengths (including/excluding the RGB image) were determined by using a piecewise selection method. Validation tests were carried out on 3636 images of the food types that were used in training the CNN. As a result of the experiments, the accuracy of food classification was increased from 88.9 to 97.1% and MAPEs were decreased from 41.97 to 18.97 even when one kind of NIR image was added to the RGB image. The highest accuracy for food type classification was 99.81% when using 19 images and the lowest MAPE for caloric content was 10.56 when using 14 images. These results demonstrated that the use of the images captured at various wavelengths in the UV and NIR bands was very helpful for improving the accuracy of food classification and caloric estimation.

## 1. Introduction

Precise and continuous monitoring of the types and amounts of foods consumed is very helpful for the maintenance of good health. For health professionals, being aware of the nutritional content of ingested food plays an important role in the proper treatment of patients with weight-related diseases and gastrointestinal diseases as well as those at high risk for metabolic diseases such as obesity [1]. For people with no health problems, monitoring the types, amounts, and nutritional content of food consumed is useful in order to maintain this status. Monitoring of the types and amounts of foods eaten is often achieved via manual record-keeping methods that include food-frequency questionnaires [2], self-report diaries [3], and multimedia diaries [4]. Several user-friendly diet-related apps have become available on smartphones in which image recognition schemes are in part adopted to classify the types of food. In such an approach, however, the accuracy is affected by user inattention and erroneous record-keeping that often decreases usefulness.

Several automatic food recognizers (AFRs) are available to continuously recognize the types and amounts of consumed food with minimum user intervention required. Wearable sensing and digital signal-processing technologies are key factors in the implementation of AFRs which are divided into several categories according to the adopted sensing method. In acoustic-based methods, classifying the types of food is achieved via chewing and swallowing sounds. The underlying principle is that chewing sounds vary depending on the physical characteristics of the food, which includes shape, hardness, and moisture content. In-ear microphone [5,6,7] and throat microphone [8,9] are typically used to acquire the sounds of food intake. By using a throat microphone, recognition experiments were carried out on seven types of food [9]. A recognition rate of 81.5∼90.1% was achieved where a hidden Markov model (HMM) was adopted to classify the types of food [9]. Päfiler et al. performed recognition experiments on seven types of food using an in-ear microphone and reported a recognition rate of 79∼66%. The performance achieved using acoustic signals has been limited, however, because it is difficult to discriminate various foods by using only acoustic cues.

A variety of sensors are used in non-acoustic methods. These include an imaging sensor [10], a visible light spectrometer, a conductive sensor [11], a surface electromyogram (sEMG) sensor attached to the frame of eyeglasses [12], and a near-infrared (NIR) spectrometer [13]. These methods have the advantages of distinguishing and sub-dividing different types of food while analyzing the principle constituents. Sensors that are inconvenient to wear, however, can be disadvantageous and separate sampling of the food is required [11,13]. Ultrasonic Doppler shifts are also employed to classify the types of food [14]. The underlying principle of non-acoustic methods involves movements of the jaw during the chewing of food, as well as vibrations of the jaw caused by the crushing of food, both of which reflect the characteristics of food types. The accuracy of the ultrasonic Doppler method was 90.13% for the six types of food [14].

Since types of food are easily distinguished according to their shape, texture, and color, visual cues have been used for the classification of food types and estimation of the food amount [10,15,16,17,18,19,20,21,22,23]. In a vision-based approach, the classification of food types can be formulated as pattern recognition problems where segmentation, feature selection and classification are sequentially carried out for food images. Due to recent advances in machine learning technology, an artificial neural network has been employed to classify food categories and to predict the caloric content of food [18,22,23]. Convolutional neural networks (CNNs) have been used to established 15 food categories with an average classification accuracy of 82.5% and a correlation between the true and estimated calories of 0.81 [18]. When RGB images under visible lighting sources were used in previous vision-based approaches, recognition accuracy was degraded for visually similar foods. Moreover, the lack of a specific reaction to UV and NIR lights is a limitation meaning that this technology cannot be utilized in food analysis.

Multispectral analysis has been widely adopted in food analysis [24,25,26,27,28,29,30,31,32,33,34,35,36,37,38,39]. The underlying principle is that individual ingredients in food have different absorption spectra. For example, infrared (IR) light is strongly absorbed by water compared with ultraviolet (UV) and visible (VIS) light. Therefore, differences in the absorption spectra between the VIS and IR bands are useful in estimating the amount of water contained in food. In previous studies, multispectral analysis of food was employed to quantify specific contents, such as oil, vinegar [24], water [27], sugar [28,29,30,31,32,33,34,35,36], and soluble protein [37]. The multispectral analysis involved the use of a spectrometer and a wide-band light source (such as a halogen lamp). Using these methods, an optimal set of wavelengths was chosen from the absorption spectra in the interest of maximizing the prediction accuracy for the ingredients of interest. A correlation of 0.8912 was obtained using four wavelengths out of a 280-bin absorption spectrum when predicting the sugar content of apples [28]. When hyperspectral imaging was adopted to predict the sugar content in wine, a maximum correlation of 0.93 was obtained when using partial least squares regression [32]. The usage of a spectrometer and a wide-band light source confers the ability to select the optimal wavelength in sharp detail. Problems associated with size, weight, and power consumption, however, could potentially cause difficulties in implementing wearable monitoring devices.

The multispectral approach has also been implemented by using a number of narrow-bandwidth light sources, such as light emitting diodes (LEDs), and a digital camera [24,26,27,31,35,39]. Compared with halogen lamps, the improvement gained when using an LED light source was verified for multispectral food analysis [31]. Experimental results obtained showed that use of an LED light source returned a slightly higher correlation than that of halogen lamps (0.78 vs. 0.77). The number of employed wavelengths ranged from 5 [39] to 28 for UV, VIS, and NIR bands [31]. Raju et al. [24] used the multispectral images from 10 LEDs with different wavelengths to detect dressing oil and vinegar on salad leaves, and reported an accuracy of 84.2% by using five LEDs. Previous studies were focused mainly on predicting the specific nutritional content of specific foods (e.g., water in beef [27], sugar in apples [28], sugar in sugarcane [29], sugar in peaches [30], soluble solids in pomegranates [33], sugar in potatoes [34], sugar in black tea [35], and soluble protein in oilseed rape leaves [37]). The caloric content of food is determined by the amounts of each ingredient, and it would be reasonable to expect accurate predictions when using multispectral analysis techniques.

In the present study, multi-wavelength imaging techniques were applied to classifying food items and estimating caloric content. Compared with conventional RGB image-based methods, the usefulness of NIR/UV images was experimentally verified for food classification and predictions of caloric content. The optimal number and combination of the wavelengths was determined to maximize the estimation accuracy where a piecewise selection method was adopted.

This paper is organized as follows: In Section 2, the processing of data preparation and the properties of the employed food items are presented. The preliminary verification results and the overall procedure for food analysis are explained in Section 3. The experimental results and discussion of the results are presented in Section 4. Concluding remarks are provided in Section 5.

## 2. Data Acquisition

The list of the food items used in this study is presented in Table 1. The food items were selected to represent the various physical properties (liquid/soft/hard) of everyday foods and to reflect the naturally occurring balances between healthy and unhealthy foods. The caloric amount was obtained from existing nutrition fact tables for each food and food composition data released by the Ministry of Korea Food and Drug Safety (KFDA) [40]. It is noteworthy that a number of foods were nutritionally different but were difficult to distinguish visually (e.g., cider and water, coffee and coffee with sugar, tofu and milk pudding, milk soda and milk…). Such pairs of food items were good choices for verifying the feasibility of UV and NIR images in the recognition of the types and calories of food. In the case of liquid food, images were acquired by putting the same amount of food in the same container (cup) so that the shape of the cup and the amount of food were not used as a clue for food recognition. In a similar manner, in the case of non-liquid foods, plates of the same size and shape were used.

A custom-made image acquisition system was employed to obtain the multispectral images. A schematic of the image acquisition system appears in Figure 1. The photography is shown in Figure 2. The light source was positioned 25 cm above the food tray. Four digital cameras faced the center of the food tray and were connected to the desktop PC via a universal serial bus (USB). The acquisition image size was set at 640 × 480 pixels (HV), and each pixel had a 16-bit resolution. Each camera was equipped with a motorized IR-cut filter to block the visible lights when the IR images were acquired. The light source consisted of a total of 20 types of LEDs emitting light at different wavelengths (385, 405, 430, 470, 490, 510, 560, 590, 625, 645, 660, 810, 850, 870, 890, 910, 950, 970, 1020 nm, and white). The white LED was used to obtain the RGB images, which were split into three (R-G-B) channels. The light source of each wavelength was composed of 30 LEDs with the exception of the white light source(10 LEDs). The LEDs of each wavelength were arranged in a circular shape at a specified position on the printed circuit board (PCB), as shown in Figure 3. Before the image of a specific wavelength was acquired, the center of the corresponding LED area was moved to the center of the food tray. Since the intensities of LEDs were different according to wavelength, the driving current of LEDs for each wavelength was adjusted to minimize the differences in light intensity according to the wavelength. The LED panel moved back-and-forth and left-to-right using the linear stages powered by the stepping motors. Data augmentation was achieved not by image transformations but by capturing images from as many views as possible. Accordingly, the four cameras and a rotating table were employed. The angular resolution of the rotating table could be adjusted from 0.5∼90${}^{\circ }$. The movement of the LED panel, rotation of the tables, and the on/off switch of each of the LEDs were all controlled by a micro-controller (Atmega128A).

The number of control commands was predefined both in the control module of the acquisition system and in the host desktop PC. Hence, the task of acquisition was achieved entirely by constructing the sequence of the individual commands. The acquisition code was written in Python (version 3.6.11). The communication between the desktop PC and the control module was achieved using Bluetooth technology. Image acquisition was carried out in a dark chamber (470 mm × 620 mm × 340 mm, WDH) where external light was blocked. The angular resolution of the rotating table was set at 10${}^{\circ }$ (total 36 views per camera). The total acquisition time for each food was 2738 s, which corresponded to an acquisition time of 3.8 s per frame. The images of bread, castella, and a chocolate bar were captured under white light from specific angles, as shown in Figure 4.

## 3. Food Analysis

### 3.1. Preliminary Feasibility Tests for UV/NIR Images

The main objective of this study was to improve the accuracy of food classification and calorie estimation by using UV/NIR images. Prior to construction of the classification/estimation rules, the use of UV/NIR images was experimentally verified as adequate for this purpose. The white LED used for capturing RGB images was experimentally measured as emitting light with a wavelength that ranged from 430 to 640 nm. Hence, images acquired under a light source with a wavelength shorter than 430 nm were considered UV images, and the images acquired under a light source with a wavelength longer than 640 nm were considered NIR images. Two food items could not be well-distinguished visually under a visible light source, which necessitated confirming whether the corresponding images acquired under UV or NIR light sources could be relatively well-distinguished. Food images are visually distinguished according to shape, texture, and the distribution of brightness values (histogram). The shapes and textures of foods, however, are uniquely determined independent of the light source, whereas the distribution of brightness values is affected not only by food types but also by the wavelength of the light source. Accordingly, brightness distribution was used to measure the difference between two food images that are affected by the wavelength of the light source. A histogram was obtained from the image in which the non-food portion was masked. In the present study, the Bhattacharyya distance was employed to measure the differences between two food images. The Bhattacharyya distance for two images $I_{n}$ and $I_{m}$ acquired under a light source of a wavelength $\lambda_{k}$ is given by
(1)$$D_{B}\left(I_{n},I_{m}|\lambda_{k}\right)=-\log\{\sum_{y\in Y}\sqrt{p\left(y|I_{n},\lambda_{k}\right)p\left(y|I_{m},\lambda_{k}\right)}\}$$
where *Y* is the set of possible brightness values, and p(y|I,λ) is the probability density function of the brightness value *y* included in image *I* at a wavelength of λ. The Bhattacharyya distance represents a complete match at a value of 0 (minimum) and a complete mismatch at a value of 1 (maximum). Figure 4 presents an example of the two different food images acquired under the different wavelength light sources. Two food items, coke and sugar-free grape juice looked similar under a visible light source, as shown in the first row of Figure 4, whereas the differences between these two food images were apparent in the IR images at 810 nm. For this image pair, the Bhattacharyya distances for λ=640nm and λ=810nm were given as 0.45 and 0.99, respectively. This example shows that the Bhattacharyya distance can be a good indicator of visual differences between two different food items.

The usefulness of UV/NIR images in terms of food classification was verified by examining the proportion of food pairs that were visually similar under visible light but visually distinct under UV or NIR light. To this end, a cumulative distribution function was defined; it can be heuristically computed as follows:(2)$$F_{C}\left(T_{V},T_{NV}\right)\approx \frac{\left|\{\left(I_{m},I_{n}\right)|D_{B}^{\left(V\right)}\left(I_{n},I_{m}\right)<T_{V}\;\mathrm{and}\;D_{B}^{\left(NV\right)}\left(I_{n},I_{m}\right)>T_{NV}\}\right|}{\left|\{\left(m,n\right)|D_{B}^{\left(V\right)}\left(I_{n},I_{m}\right)<T_{V}\}\right|}$$
where $\left(I_{m},I_{n}\right)$ is the pair of two different food items *m* and *n*, and |S| is the cardinal of the set *S*. In this study, the maximum $D_{B}$ value in each band (visible and non-visible bands) was chosen as the representative $D_{B}$ for the corresponding band, and, hence, $D_{B}^{\left(V\right)}$ and $D_{B}^{\left(NV\right)}$ are given by
(3)$$\begin{array}{ccc} D_{B}^{\left(V\right)}\left(I_{n},I_{m}\right) & = & \max_{\lambda \in \Lambda_{V}}D_{B}\left(I_{n},I_{m}|\lambda \right)\\ D_{B}^{\left(NV\right)}\left(I_{n},I_{m}\right) & = & \max_{\lambda \in \Lambda_{NV}}D_{B}\left(I_{n},I_{m}|\lambda \right) \end{array}$$
where $\Lambda_{V}$ and $\Lambda_{NV}$ are the sets of visible and non-visible wavelengths, respectively. $T_{V}$ and $T_{NV}$ are the thresholds of the Bhattacharyya distances for the visible band and non-visible band, respectively, and $F_{C}\left(T_{V},T_{NV}\right)$ represents the ratio of image pairs that satisfy the condition whereby the Bhattacharyya distance from the VIS images is less than $T_{V}$ while the Bhattacharyya distance from UV or IR images is greater than $T_{NV}$. The case of $T_{V}<T_{NV}$, $F(T_{V},T_{NV})$ represents the frequency of a relatively small Bhattacharyya distance (a low degree of discrimination) in the visible light band but a high Bhattacharyya distance (a high degree of discrimination) in the non-visible light band.

The cumulative distributions for the various thresholds are plotted in Figure 5. Note that the visual difference between two images was not significant at a Bhattacharyya distance of 0.45, as shown in Figure 6. And, hence, the curves with $T_{V}\leq 0.4$ correspond to the cumulative distribution obtained from food pairs that are not well distinguished under a visible light source. In the case of $T_{V}=0.4$, the ratio of the Bhattacharyya distance for UV/NIR images exceeding 0.5 (corresponds to well-distinguished) was about 70%. Similar results were obtained for other $T_{V}$ values. (e.g., For $T_{V}$ = 0.2, 0.3, and 0.4, $F_{C}\left(T_{V},T_{NV}=T_{V}\right)$ = 89, 77, and 64%, respectively) This means that a significant number of food pairs that were not visually well-distinguished under visible light sources was better distinguished under non-visible light sources. Such results indicate that the performance of food classification can be improved by using UV/NIR images that are complementary to VIS images.

In terms of caloric estimation, the usefulness of a specific wavelength image was determined by examining whether differences in the amount of calories between two food items were significantly correlated with differences between the two corresponding images. The underlying assumption is that if the difference between the two food images is large, their caloric difference will also be large and vice-versa. The caloric count was computed via the measured weight and the nutrition facts for each food. The Bhattacharyya distances were also adopted to measure the differences between the two food items. The caloric difference between the two food items *n* and *m* is given by the following absolute relative difference:(4)$$D_{c}\left(n,m\right)=\frac{|c_{n}-c_{m}|}{c_{n}+c_{m}}$$
where $c_{n}$ and $c_{m}$ are the calories of food items *n* and *m*, respectively. The Pearson correlation for the *k*-th wavelength images is given by
(5)$$\rho \left(\lambda_{k}\right)=\frac{\mathrm{cov}[D_{B}\left(I|\lambda_{k}\right),D_{C}]}{\sigma_{D_{B}\left(I|\lambda_{k}\right)}\sigma_{D_{C}}}$$
where cov[x,y] denotes the covariance of *x* and *y* and $\sigma_{x}$ is the standard deviation of *x*. $D_{B}\left(I|\lambda_{k}\right)$ denotes the Bhattacharyya distance computed from the images acquired under a light source of $\lambda_{k}$ wavelength.

The correlations are presented across the wavelengths of each light source in Figure 7. The maximum correlation was obtained at λ=870nm. A second maximum also appeared in the wavelength of the NIR band (λ=850nm). The results indicate that the differences between the NIR images are moderately correlated with differences in caloric content. The images acquired under the NIR light source are beneficial in terms of caloric estimation. The average correlations of the visible and non-visible bands were 0.636 and 0.633, respectively. The significance test also showed that there was no remarkable difference between the correlation values of the visible band and those of the non-visible band (*p* = 0.7). From such results, it can be reasonably assumed that VIS- and UV/NIR-images are equally useful in terms of caloric estimation.

### 3.2. Preprocessing

Although a highly stable current source was adopted to drive the LEDs, there was some variation in the intensity of the light from shot to shot. This caused unwanted changes in the acquired images and resulted in degradation of the estimation accuracy. A simple way to compensate for variations in the intensity of light sources is to adjust the intensity of the incident light so that the average intensity approximates that of the reference intensity. A typical scale factor is given by
(6)$$\alpha_{\omega }^{\left(i\right)}=\frac{\mu_{\omega }^{\left(i\right)}}{\mu_{\omega }^{\left(ref\right)}}$$
where ω and *i* are the indices of the wavelength and shot, respectively, and $\mu_{\omega }^{\left(i\right)}$ and $\mu_{\omega }^{\left(ref\right)}$ represent the average and reference intensities, respectively. The reference intensity can be obtained by averaging a large number of light source images at different times. Such an intensity normalization method is very simple and easy to implement.

### 3.3. Food Analysis Using a Convolutional Neural Network

There are many machine learning schemes, such as random forest (RF), support vector regression (SVR), partial least squares regression (PLSR) and artificial neural networks (ANNs), that can be applied to recognition of the types, amounts, and nutritional content of food. Among them, the ANN-based approaches have an advantage wherein nonlinear relationships between the input (multispectral images) and the output (target values) can be taken into consideration in constructing the estimation rules. This leads to higher performance in terms of classification and regression. Accordingly, a supervised learning approach that employs convolutional neural networks (CNNs) was adopted in this study. The architecture of the CNN adopted in this study is shown in Figure 8, and was heuristically determined using a validation dataset (10% of the entire learning dataset). Note that the CNN architecture shown in Figure 8 was used to classify the types of food. The final output was replaced by a single output in the case of caloric estimation. The resultant architecture of the CNN was simpler than others previously proposed in image recognition tasks (e.g., ResNet-50, Inception-v3, Xception). This was due mainly to the smaller number of targets compared with that of other CNNs (101 vs. 1000).

We tested the performance in terms of food classification accuracy and estimation error for target values according to different sizes of CNN input (input image sizes). The results showed that sizes of 64 × 64 yielded the highest performance for both classification and estimation accuracies. Accordingly, all images from the camera were reduced to 64 × 64 by using interpolation. Before reducing the size, no cropping was carried out on the acquired images, and the entire image size (640 × 480) was used.

The performance according to the hyperparameters of CNN was also investigated using a validation dataset. This was performed separately for each task (food classification and caloric estimation). The resultant CNN was composed of two convolution/max pooling layers and a fully connected multi-layer perceptron (MLP) with a recognition output, as shown in Figure 8. The kernel sizes of the first and the second convolution layers were 11 × 11 and 7 × 7, respectively, while the window sizes of the max pooling layers were commonly 4 × 4. There were three fully connected layers in the employed CNN, which corresponded to input from the final convolution layer, and to the hidden and output layers. The numbers of nodes for each of the layers were also determined using the validation datasets 112, 128, and 18, respectively. Although the hyperparameters of each CNN were separately tuned for each task, the architecture of the CNN, shown in Figure 8, yielded satisfactory performance for both image classification and caloric estimation.

A rectified linear unit (ReLU) was adopted as an activation function for all hidden layers. A soft-max function and linear combination function were employed for the output layer for classification and regression CNNs, respectively. Accordingly, the loss functions were given by the cross-entropy and the mean absolute percentage error (MAPE) in the cases of food classification and caloric estimation, respectively. A total of 1000 epochs resulted in a trained CNN with sufficient performance in terms of food recognition accuracy. It is noteworthy that the accuracy of classification/estimation was strongly affected by the mini-batch size. The experimental results showed that a mini-batch size of 32 gave the best performance for all cases. Since the MAPE is given by dividing the absolute error value by its ground truth, the loss value cannot be calculated when the given target value is zero. As shown in Table 1, there were some cases when the ground-truth caloric count was zero. Note that a value of zero in the nutrition facts does not necessarily mean that the amount of the nutritional content is zero. A value of zero actually means that the amount is less than its predefined minimum. In the present study, zero caloric values were replaced by the minimum, which was 5 (kcal) according to [40].

### 3.4. Selection of the Wavelengths

Although evaluation of all possible wavelength combinations was performed via off-line processing, it was desirable to avoid an enormous amount of computational time for a brute-force grid search. In the present study, a piecewise selection method was adopted to select the set of optimal wavelengths. Let $\Omega =\{\omega_{0},\omega_{1},\ldots ,\;\omega_{N-1}\}$ be the set of the employed wavelengths, where *N* is the total number of the wavelengths. The set of the wavelengths was gradually constructed by adding and removing the wavelength either to or from the previously constructed set. The overall procedure is as follows:

Step-1. Forward selection: Let $\Omega_{F}^{\left(i\right)}$ be the set of the wavelength at the *i*-th forward step, all combinations $\Omega^{\left(i-1\right)}\cup \left\{\omega |\;\omega \in {\bar{\Omega }}^{\left(i-1\right)}=\Omega -\Omega^{\left(i-1\right)}\right\}$ are evaluated to find the optimal wavelength $\omega_{F}^{*}$ that minimizes the given loss function, then construct $\Omega_{F}^{\left(i\right)}=\Omega^{\left(i-1\right)}\cup \left\{\omega_{F}^{*}\right\}$.

Step-2. Backward elimination: The element (wavelength) that minimizes the loss function is removed from $\Omega_{F}^{\left(i\right)}$. The set of the wavelength at the *i*-th backward step is then given by
(7)$$\Omega_{B}^{\left(i\right)}=\Omega_{F}^{\left(i\right)}-\left\{\omega_{B}^{*}\right\}$$
where
$$\omega_{B}^{*}=\arg\min_{\omega \in \Omega_{F}^{\left(i\right)}}L\left(\Omega_{F}^{\left(i\right)}-\left\{\omega \right\}\right)$$
and L(S) is the loss for a wavelength set *S* that is given by the final loss of the learned CNN.

Step-3. Final forward selection: The final set is built by the forward selection step wherein the optimal wavelength $\omega^{*}$ is chosen from the set $\Omega -\Omega_{B}^{\left(i\right)}$ so as to minimize the loss function.

The procedure Steps-1∼3 was iteratively performed until $\Omega^{\left(i\right)}=\Omega$. The final set of the optimal wavelengths is given by
(8)$$\Omega^{*}=\min_{i}L\left(\Omega^{\left(i\right)}\right)$$

A large number of the wavelengths in $\Omega^{*}$ involved an increased number of LEDs and shots, which resulted in a large device, a high rate of power consumption, and long acquisition times. Hence, it is preferred that the number of wavelengths should also be considered in building the estimation rules. The three-channel (RGB) images could be obtained from one image acquired under the white light that could be regarded as a representative image within the visible light region. In the present study, food analysis was performed in combination with RGB, UV, and NIR images to reduce the number of shots and the results were compared by combining all the images acquired in the full set of wavelengths.

## 4. Experimental Results

### 4.1. Accuracy for Food Item Classification

The food classification accuracy according to the number of input images is presented in Figure 9. The number of input images is equal to the number of images actually taken by the camera at a different wavelength. Note that although the RGB image was separated into three individual images that were inputted to CNN, they were considered as one image because they were captured by one shot. The results indicated that the accuracy was increased rapidly when the number of images was less than five. Even when one image was used, the classification accuracy was higher than the previous CNN-based food recognition method [18]. Such a high level of accuracy was due mainly to the usage of a large-sized training dataset that included images acquired from various directions. A maximum accuracy of 99.81% was obtained when images acquired from 11 different wavelength light sources were used. However, the accuracy increased until the number of images was five, at which point no significant increase was observed when the number of images exceeded five. The maximum increase in classification accuracy (7.67%) was obtained when the number of input images was increased from one to two. This was confirmed by the fact that the correlation coefficient between the number of images and classification accuracies was 0.919 for as many as five images. The correlation coefficient was decreased to 0.587 when the number of images ranged from 6 to 19. Such results indicate that it is possible to obtain a sufficient recognition rate by using five images at different wavelengths. In fact, the recognition rate obtained from the five images was 99.12%, which was not significantly lower than the maximum recognition rate of 99.81%.

Thus far, the results were obtained by excluding RGB images. The food classification accuracy is presented in Figure 9, where single-wavelength images were added to the RGB image. The increase in recognition rate was more rapid compared with when the RGB image was not used. A maximum recognition rate of 99.83% was obtained when almost all images (19 out of 20) were used. The largest increase in classification accuracy was achieved when only one single wavelength image was added to the RGB image (e.g., accuracy was increased from 88.86 to 97.08% when an image from a wavelength of 890 nm was added to the RGB image.) The accuracy gradually increased until the number of images added exceeded three, and remained almost constant until the number exceeded five. When food classification was performed with only one type of image (as with the previous image-based food classification methods), the correct recognition rates of 88.86% and 87.9% were achieved by using the RGB image and one other image from a single wavelength, respectively. This indicates that an RGB image is a slightly better choice for food classification when only one type of image is used.

As noted in the previous section, a high level of accuracy is paramount when using a small number of images. From this point of view, it is meaningful to examine how the correct recognition rate changes according to the food item when recognition is performed by adding only one type of NIR or UV image to an RGB image. The experimental results on two input images (including the RGB image) showed that the highest classification accuracy was achieved when an image from a wavelength of 890 nm was added to the RGB image. Accordingly, the change in the recognition rate for each food item was examined when adding only one NIR image from a wavelength of 890nm to the RGB image. From among 101 food items, 62 items showed an improved recognition rate by adding only one type of NIR image, and 35 items showed the same recognition rate. A decrease in the recognition rate was observed for only four food items, but the level of decrease was generally small (<5%). The list of food items that resulted in a significant level of improvement in the recognition rate by adding one type of NIR image is presented in Table 2. It is noteworthy that, for most of these food items, there exist other food items that are not easily distinguished under visible light. For example, the food pairs coffee and coffee with sugar, as well as caffelatte and caffelatte with sugar, were almost visually identical. These food items were often recognized as other foods that appeared almost identical under visible light. For the food item of grape soda, a recognition rate of 0% was obtained when only an RGB image captured under visible light was used. In this case, all grape soda images were recognized as sugar-free grape soda that is visually identical to grape soda. When adding an image acquired under a 890 nm wavelength light source, a recognition rate of 44% was obtained. Consequently, it is apparent that the NIR/UV images improve the accuracy of image-based food classification.

### 4.2. Accuracy for Caloric Estimation

The results for caloric estimation are presented in Figure 10 in which MAPEs are plotted for each number of images actually taken. Without an RGB image, the minimum of MAPE was 10.49 when a total of 10 different wavelength images were employed. Interestingly, MAPE was decreased until the number of input images reached 10 ($R^{2}=-0.9317$), but was increased after the number of input images exceeded 11 ($R^{2}=0.6530$). That result was likely due to the limitations of the piecewise algorithm adopted in the selection of the wavelengths and overfitting problems caused by an excessive increase in the number of input images. When the three wavelength images were employed, the MAPE was 18.97%, which was significantly lower than when using the RGB image alone (=41.97%). Such a result was also remarkably better than those of the previous CNN-based caloric estimation schemes (27.93% [19], even though the number of the food items adopted was relatively large (101 vs. 15). The selected wavelengths were 385, 560, and 970 nm in the case of three input images, indicating that one each of UV, VIS, and NIR wavelength band images was selected. This implies that not only the number of input images, but also the selected wavelengths play an important role in the accuracy of caloric estimation.

When single-wavelength images were added to an RGB image, a minimum of 10.56% MAPE was achieved. A total of 14 input images were needed to obtain the minimum MAPE, which meant more images were required compared with when the RGB image was excluded (10 images). Overall, the MAPEs take the form of decreasing curves with an increasing number of input images ($R^{2}=-0.5935$). However, the correlation values before and after the minimum point (14) were −0.6847 and −0.4919, respectively, indicating that there was no significant difference between the number of input images and MAPE when the number of input images exceeded 14. The Pearson correlation for the caloric counts between the ground truth and the estimation was also notably larger than that of the previous CNN-based estimation schemes [18,19] (0.975 vs. 0.806) when a total of 14 UV/VIS/NIR images was employed in the caloric estimation. This was due mainly to the usage of the UV/NIR images in this study. Even for visually similar foods with different caloric counts, their UV/NIR images were often clearly distinguishable. Adding more images obtained from light sources with various wavelengths in the UV and NIR bands progressively improved the process of classifying images according to the caloric count.

Figure 11 compares the cumulative distribution function of MAPE when using only RGB images to also using other images acquired from a single wavelength light source. The superiority of using additional images along with RGB images is confirmed by this figure. For example, about 50% of all test food images revealed a MAPE value of less than 30 in the case of using only RGB images. When adding one image to the RGB image, 50% of all test food images showed a MAPE value of less than 12. This was further reduced to five when five images were added to the RGB image.

It was also important to examine the accuracy of the caloric estimation for each food item when adding only one type of NIR/UV image to the RGB image. An image acquired at a wavelength of 970 nm represented the highest MAPE reduction rate and was chosen as an addition to the RGB image. Compared with the use of only RGB images, the number of food items with decreased, increased, and maintained MAPE values was 77, 20, and 1, respectively (out of 98 food items with valid caloric values). This indicates that the accuracy of caloric estimation for many food items (>78%) can be improved by including only one type of NIR image. This is also confirmed by the fact that most of the MAPE reduction (74.8%) was achieved when only one NIR image (at 970 nm) was added to the RGB image, as shown in Figure 11. Table 3 lists the large differences in MAPE when only the RGB image was used compared with using the RGB + 970 nm images. The estimated caloric values of the food items presented in Table 3 deviated from the ground truth by more than 50%. The results show that the MAPEs values for these food items were reduced by more than half when calories were estimated after adding one type of NIR image to the RGB image.

In conclusion, the use of UV/NIR images in additionj to RGB images increases the accuracy of caloric estimation. Even with a smaller number of UV/NIR images, the performance in terms of caloric estimation was significantly improved compared with conventional RGB-based estimation.

### 4.3. Analysis of the Selected Wavelengths

The wavelengths selected for food classification appear in Table 4 for each number of images. The selection rate for each wavelength is shown in Figure 12. The results are presented for two cases: inclusion and exclusion of RGB images. When the RGB image was excluded, the ratio of wavelengths corresponding to the NIR band being selected was 46.84%. On the other hand, 41.05% of wavelengths corresponded to the visible light band. The UV band was selected at a relatively low rate (12.11%). This was partially because the number of employed UV light sources was smaller than that of the UV/VIS light sources. Hence, the light source in the UV band was not likely to be selected. The chances of selecting the wavelength of the NIR band was increased to 60% in cases when a relatively small number of wavelengths was allowed (≤5). Considering the fact that the number of wavelengths belonging to the NIR band was slightly less than the number of VIS wavelengths (8 vs. 9), a higher selection rate of wavelengths in the NIR band indicates that the NIR images are more useful in food classification compared with the VIS images. This seems to be due in part to the nature of the NIR images, where the distribution of water content in food can be approximately obtained.

When the RGB image was included, the ratio of wavelengths in the NIR band being selected was further increased to 50.53% This is because the RGB image actually includes three visible light images (RGB images), so the NIR band images are likely to be selected. Whether the RGB images were included or excluded, frequent wavelength selections were 385, 870, 890, and 970 nm. This also indicates that the wavelengths corresponding to the invisible band were more frequently selected for food classification.

The wavelength selection results for caloric estimation are shown in Figure 13 and in Table 5. When excluding the RGB images, the wavelengths in the NIR region were selected with a frequency of 31.58%, whereas wavelengths in the visible region were selected with a frequency of 51.05%. Even when RGB images were included, the wavelengths in the visible region were selected more frequently compared with those in the NIR band. This result was somewhat different from that for food classification. Wavelengths of 385, 430, 560, and 970 nm were frequently selected when excluding the RGB images, which indicates that the wavelengths corresponding to the visible band were also frequently selected. On the other hand, the wavelengths of 405, 430, 950, and 1020 nm were selected with relatively high frequency when the RGB images were included.

## 5. Conclusions

RGB images are mostly used in image-based food analysis. The proposed approach assumes that the types of food and the caloric count in them could partially be determined by the morphological characteristics and wavelength distributions of food images. A multi-spectral analysis technique using NIR and UV light along with VIS light was employed in the analysis of various foods and the results approximated those of conventional chemical analysis techniques. The present study was motivated by such a multi-spectral approach, and the procedure for food classification and caloric estimation adopted a multi-spectral analysis technique. Automated equipment capable of acquiring images of up to 20 wavelengths was devised, and approximately 15,000 images were acquired per wavelength from 101 types of food. The types of food and caloric content were estimated using a CNN. A CNN was used so that the relevant features for estimating the target variables could be automatically derived from the images, and a feature extraction step was unnecessary. The experimental results showed that the performance in terms of the accuracy for food item classification and caloric estimation was notably superior to previous methods. This was due mainly to the usage of various light sources in the UV/VIS/NIR band, unlike conventional methods which use only RGB images. It would be interesting to connect the multi-spectral imaging techniques with the quantification of ingredients such as proteins, fats, carbohydrates, sugars, and sodium. Future work will focus on this issue.

## Figures and Tables

**Figure 1 foods-12-03212-f001:**
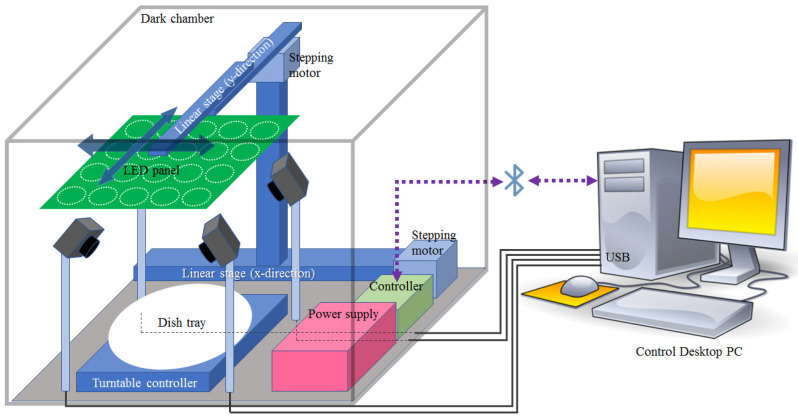
Schematic of the image acquisition system.

**Figure 2 foods-12-03212-f002:**
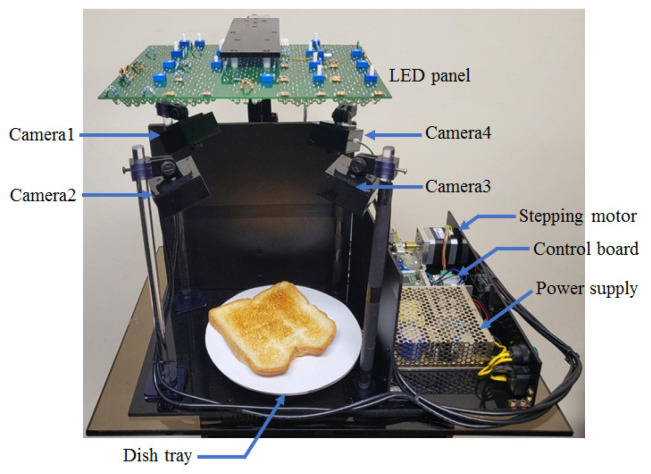
Photograph of the image acquisition system.

**Figure 3 foods-12-03212-f003:**
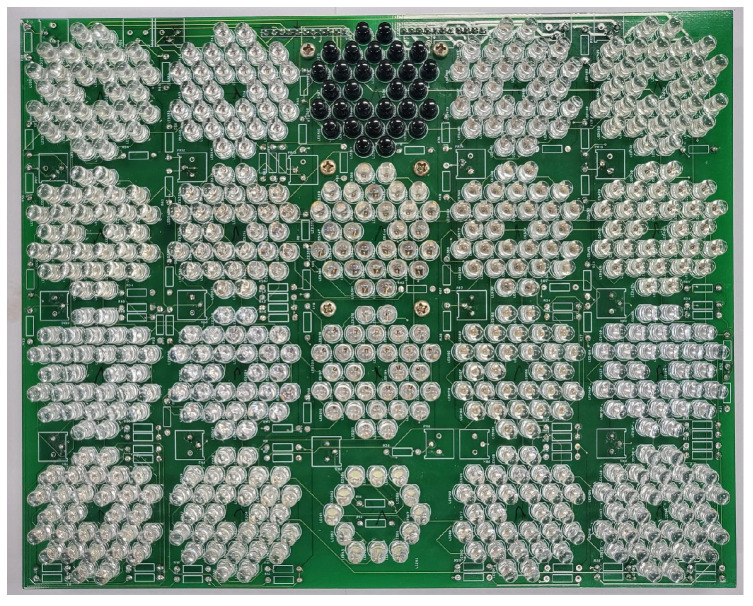
Photograph of LED in the panel.

**Figure 4 foods-12-03212-f004:**
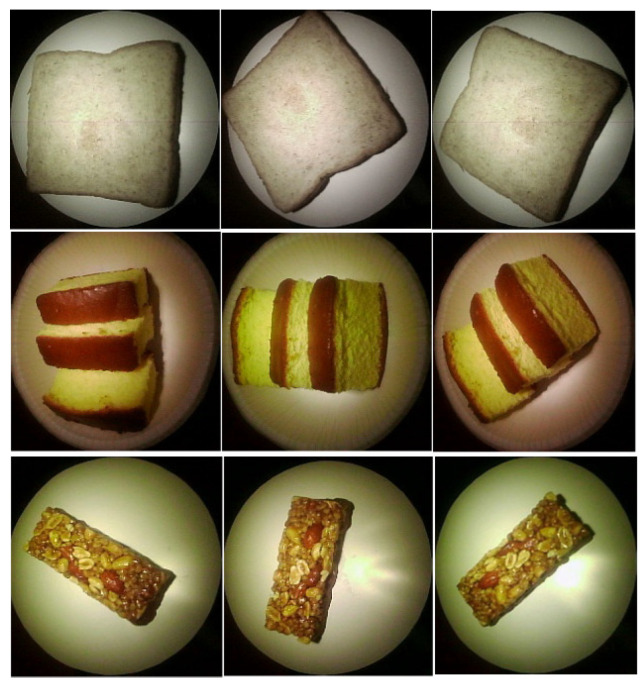
Examples of images ((**Top**): bread, (**Middle**): castella, (**Bottom**): chocolate bar) acquired from various directions.

**Figure 5 foods-12-03212-f005:**
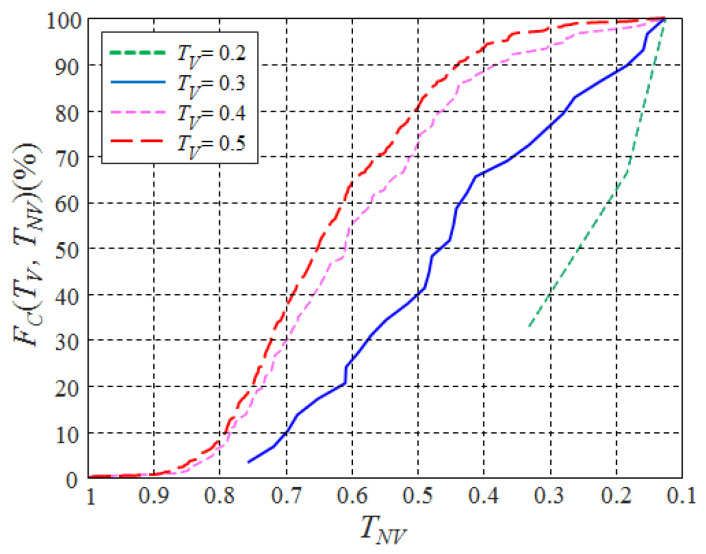
The cumulative distributions of the distances obtained from non-visible images and those from visible images.

**Figure 6 foods-12-03212-f006:**
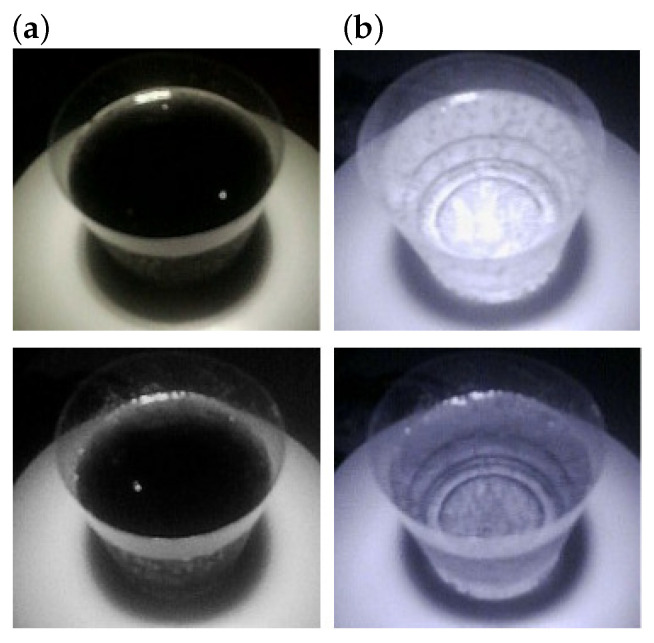
(**a**) Images acquired under a 640 nm light source. (**b**) Images acquired under a 810 nm light source. The food items are coke (**top**) and sugar-free grape juice (**bottom**).

**Figure 7 foods-12-03212-f007:**
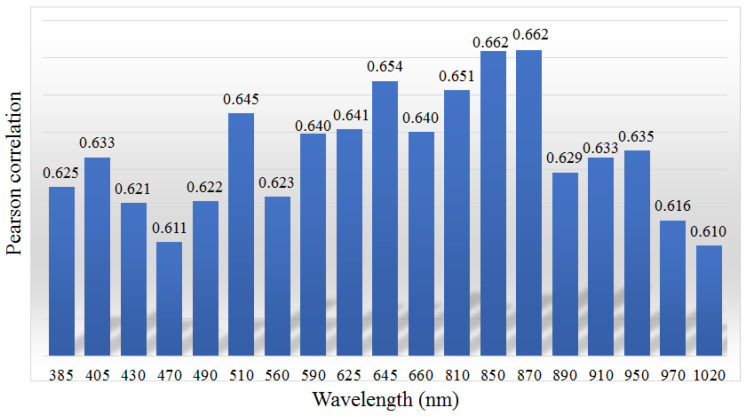
Correlations for each of the wavelength images between the caloric differences and the Bhattacharyya distances.

**Figure 8 foods-12-03212-f008:**
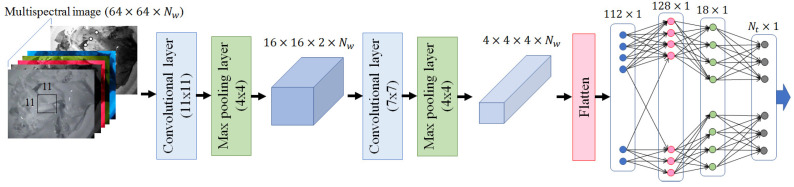
Architecture of the proposed CNN for the classification of food type, where $N_{w}$ is the number of input images and $N_{t}$ is the number of targets. (101 for food classification and 1 for caloric estimation).

**Figure 9 foods-12-03212-f009:**
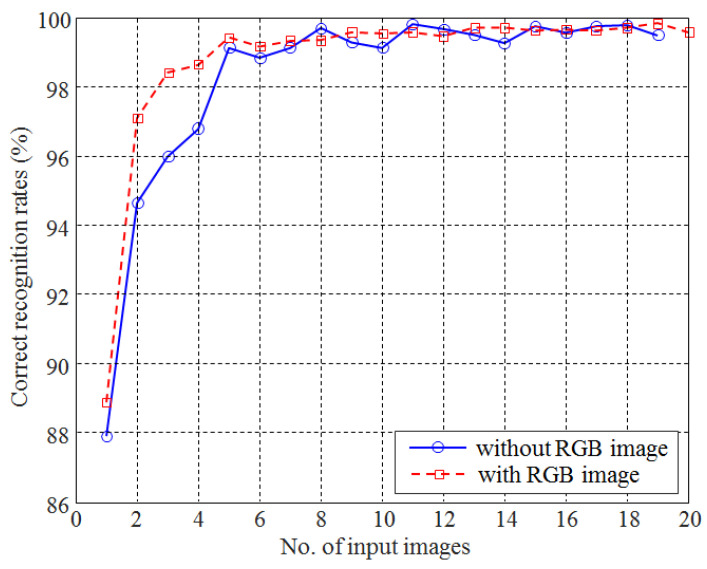
The classification accuracies of food items according to the number of light sources when the RGB image is included or excluded.

**Figure 10 foods-12-03212-f010:**
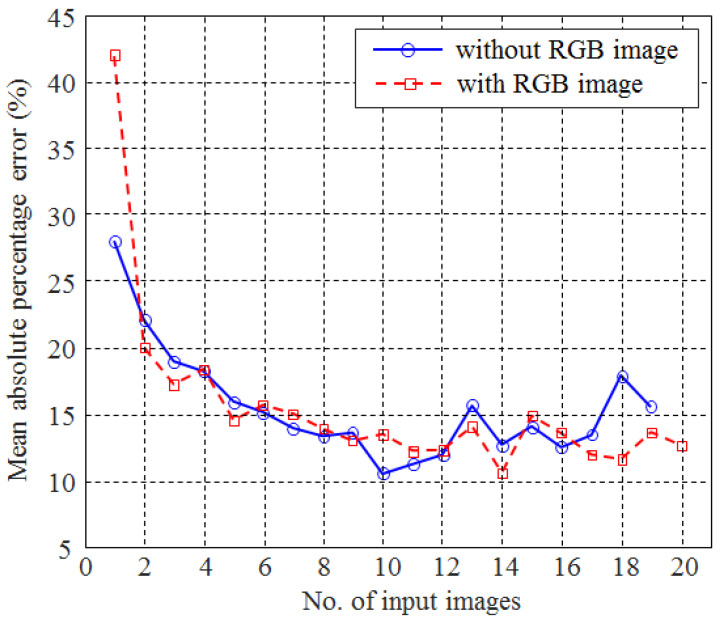
The mean absolute percentage errors (MAPEs) of caloric content (kcal) according to the number of wavelengths, when the RGB image was either included or excluded.

**Figure 11 foods-12-03212-f011:**
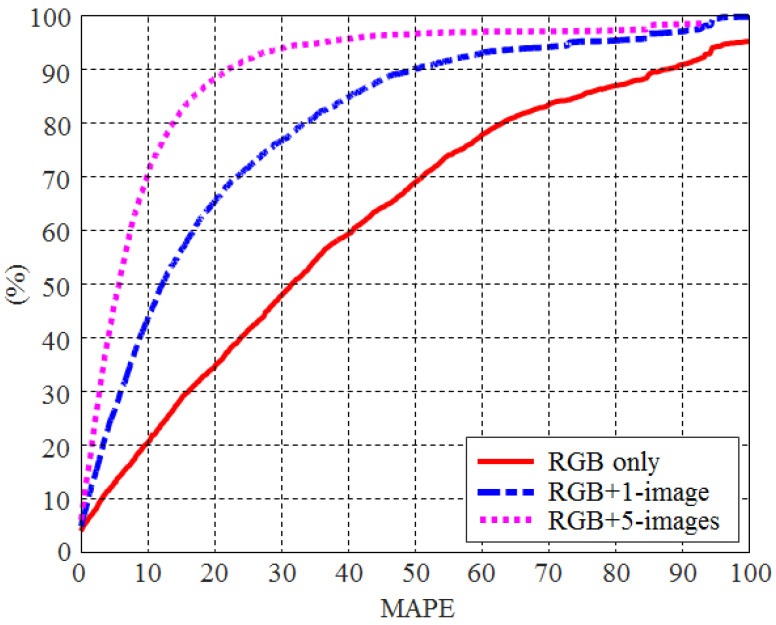
The cumulative distribution of MAPE according to the input of CNN (RGB image only, RGB + 1-images, and RGB + 5-images).

**Figure 12 foods-12-03212-f012:**
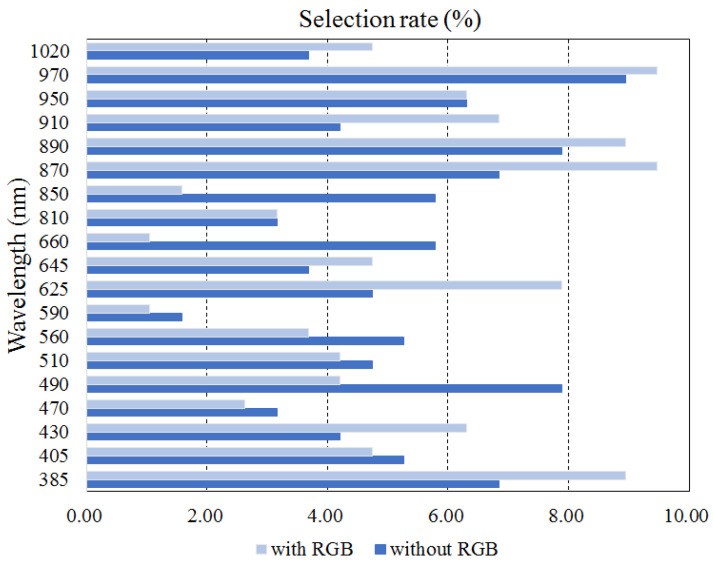
The selection ratio of each wavelength for food classification when the RGB image is included or excluded.

**Figure 13 foods-12-03212-f013:**
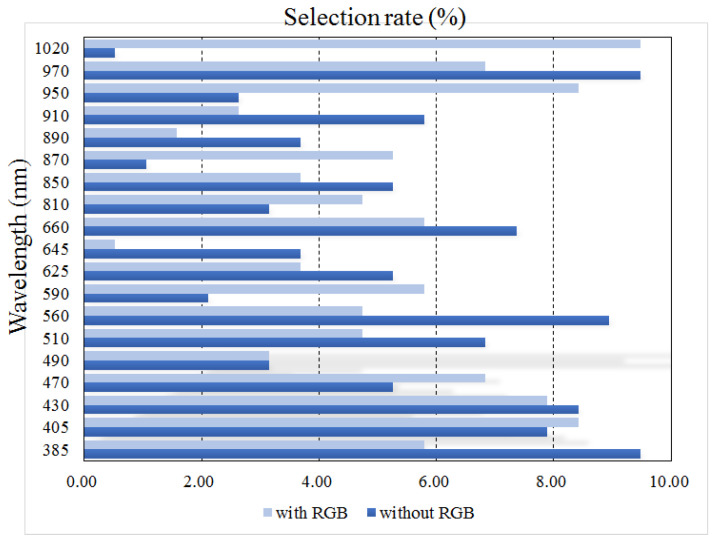
The selection ratio of each wavelength for calorie estimation when the RGB image is included or excluded.

**Table 1 foods-12-03212-t001:** Dataset properties per food item.

Food Item	Weight	Calorie	Food Item	Weight	Calorie
**Units**	**g**	**kcal**	**Units**	**g**	**kcal**
apple juice	180.5	N/a	pork (steamed)	119.3	441.41
almond milk	175.5	41.57	potato chips	23.5	130.82
banana	143.6	127.80	potato chips (onion flavor)	23.5	133.95
banana milk	174.6	110.27	sports drink (blue)	170	17.00
chocolate bar (high protein)	35	167.00	chocolate bar (with fruits)	40	170.00
beef steak	68.1	319.39	milk pudding	140	189.41
beef steak with source	79	330.29	ramen (Korean-style noodles)	308	280.00
black noodles	127.4	170.00	rice (steamed)	172.3	258.45
black noodles with oil	132.4	N/a	rice cake	119.3	262.46
blacktea	168	52.68	rice cake and honey	127.9	288.60
bread	47.8	129.54	rice juice	173.8	106.21
bread and butter	54.8	182.04	rice (steamed, low-calorie)	164.6	171.18
castella	89.9	287.68	multi-grain rice	175.3	258.08
cherryade	168	79.06	rice noodles	278	140.00
chicken breast	100.6	109.00	cracker	41.5	217.88
chicken noodles	70	255.00	salad1 (lettuce and cucumber)	96.8	24.20
black chocolate	40.37	222.04	salad1 with olive oil	106	37.69
milk chocolate	41	228.43	salad2 (cabbage and carrot)	69.1	17.28
chocolate milk	180.1	122.62	salad2 with fruit-dressing	79	28.04
cider	166	70.55	armond cereal (served with milk)	191.7	217.36
clam chowder	160	90.00	corn cereal (served with milk)	192	205.19
coffee	167	18.56	soybean milk	171.9	85.95
coffee with sugar (10%)	167	55.74	spagetti	250	373.73
coffee with sugar (20%)	167	92.92	kiwi soda (sugar-free)	166	2.34
coffee with sugar (30%)	167	130.11	tofu	138.6	62.37
coke	166	76.36	cherry tomato	200	36.00
corn milk	166.6	97.18	tomato juice	176.8	59.80
corn soup	160	85.00	cherry tomato and syrup	210	61.90
cup noodle	262.5	120.00	fruit soda	169	27.04
rice with tuna and pepper	305	418.15	vinegar	168	20.16
dietcoke	166	0.00	pure water	166	0.00
choclate bar	50	249.00	watermelon juice	177.7	79.97
roasted duck	117.2	360.98	grape soda	170.9	92.43
orange soda	173.6	33.33	grape soda (sugar-free)	170.9	0.00
orange soda (sugar-free)	166.2	2.77	fried potato	110.5	331.50
fried potato and powder	120	364.92	yogurt	179	114.56
sports drink	177.1	47.23	yogurt and sugar	144.6	106.04
ginger tea	178.3	96.79	milk soda	167	86.84
honey tea	183.9	126.69	salt crackers	41.3	218.89
caffelatte	171.6	79.13	onion soap	160	83.00
caffelatte with sugar (10%)	171.6	115.66	orange juice	182.6	82.17
caffelatte with sugar (20%)	171.6	152.19	peach (cutted)	142	55.38
caffelatte with sugar (30%)	171.6	188.72	pear juice	181.5	90.02
mango candy	36.4	91.00	peach and syrup	192	124.80
mango jelly	58.6	212.43	peanuts	37.1	217.96
milk	171	94.50	peanuts and salt	37.3	218.21
sweet milk	171	N/a	milk tea	167	63.46
green soda	174.5	84.55	pizza (beef)	85.5	212.08
pizza (seafood)	60	148.83	pizza (potato)	72.3	179.34
pizza (combination)	70.9	175.87	plain yogurt	143.7	109.89
sports drink (white)	175.8	43.95			
			mean	141.17	139.27
			standard deviation	60.69	101.36

**Table 2 foods-12-03212-t002:** Comparisons of recognition rates (%) for food items that revealed a large difference between the use of only an RGB image and using an RGB image with one additional (NIR) image.

Food Items	RGB-Image	RGB-Image
		+Image at 890 nm
black noodles with oil	58.33	100
coffee with sugar (10%)	16.67	100
orange soda	47.22	100
caffelatte with sugar (30%)	25.00	97.22
peach (cut)	55.56	97.22
milk pudding	44.44	100
rice cake	44.44	86.11
rice cake and honey	52.78	100
salad1 with olive oil	47.22	94.44
salad2 (cabbage and carrot)	47.22	94.44
grape soda	0.00	44.44

**Table 3 foods-12-03212-t003:** Comparison of MAPEs in caloric content (%) for food items showing large differences between use of the RGB-image only and use of the RGB image and one (NIR) image.

Food Items	RGB-Image	RGB-Image
		+Image at 970 nm
salad1 (lettuce and cucumber)	94.46	20.85
salad2 (cabbage and carrot)	85.2	35.37
rice with tuna and pepper	51.68	6.27
orange soda (sugar-free)	63.06	18.19
salad2 with fruit dressing	62.11	15.41
cherry tomato	58.87	21.38
fried potato and powder	55.06	25.41
green soda	52.99	17.53
salad1 with olive oil	62.65	28.58
rice cake and honey	62.1	29.52
rice cake	61.94	30.09

**Table 4 foods-12-03212-t004:** Selected wavelengths for food classification. Top: without RGB image. Bottom: with RGB image.

No. of images	Selected wavelengths (nm)																			
1	870																			
2	660	950																		
3	660	950	970																	
4	590	660	950	970																
5	490	660	890	950	970															
6	405	490	660	890	950	970														
7	385	405	490	560	660	890	970													
8	385	490	560	660	870	890	970	1020												
9	385	405	490	560	850	870	890	970	1020											
10	385	490	560	645	810	850	870	890	970	1020										
11	385	490	510	560	625	645	810	850	870	890	970									
12	385	430	490	510	560	625	810	850	870	890	910	970								
13	385	405	430	490	510	560	625	850	870	890	910	950	970							
14	385	405	430	470	490	510	560	625	850	870	890	910	950	970						
15	385	405	430	470	490	510	560	625	645	850	870	890	910	950	970					
16	385	405	430	470	490	510	625	645	660	850	870	890	910	950	970	1020				
17	385	405	430	470	490	510	625	645	660	810	850	870	890	910	950	970	1020			
18	385	405	430	470	490	510	590	625	645	660	810	850	870	890	910	950	970	1020		
19	385	405	430	470	490	510	560	590	625	645	660	810	850	870	890	910	950	970	1020	
No. of images	Selected wavelengths (nm)																			
1	RGB																			
2	RGB	890																		
3	RGB	870	970																	
4	RGB	385	870	970																
5	RGB	385	870	890	970															
6	RGB	385	625	870	890	970														
7	RGB	385	625	810	870	890	970													
8	RGB	385	625	810	870	890	910	970												
9	RGB	385	430	625	870	890	910	950	970											
10	RGB	385	430	490	625	870	890	910	950	970										
11	RGB	385	430	490	625	645	870	890	910	950	970									
12	RGB	385	405	430	625	645	870	890	910	950	970	1020								
13	RGB	385	405	430	510	625	645	870	890	910	950	970	1020							
14	RGB	385	405	430	510	560	625	645	870	890	910	950	970	1020						
15	RGB	385	405	430	490	510	560	625	645	870	890	910	950	970	1020					
16	RGB	385	405	430	470	490	510	560	625	645	870	890	910	950	970	1020				
17	RGB	385	405	430	470	490	510	560	625	645	810	870	890	910	950	970	1020			
18	RGB	385	405	430	470	490	510	560	625	645	810	850	870	890	910	950	970	1020		
19	RGB	385	405	430	470	490	510	560	590	625	660	810	850	870	890	910	950	970	1020	
20	RGB	385	405	430	470	490	510	560	590	625	645	660	810	850	870	890	910	950	970	1020

**Table 5 foods-12-03212-t005:** Selected wavelengths for caloric estimation. Top: without RGB image. Bottom: with RGB image.

No. of images	Selected wavelengths (nm)																			
1	430																			
2	385	970																		
3	385	560	970																	
4	385	430	560	970																
5	385	405	430	560	970															
6	385	405	430	560	660	970														
7	385	405	430	510	560	660	970													
8	385	405	430	510	560	625	660	970												
9	385	405	430	510	560	625	660	910	970											
10	385	405	430	470	510	560	660	850	910	970										
11	385	405	430	470	510	560	645	660	850	910	970									
12	385	405	430	470	510	560	625	645	660	850	910	970								
13	385	405	430	470	510	560	590	625	660	850	890	910	970							
14	385	405	430	470	490	510	560	625	660	810	850	890	910	970						
15	385	405	470	490	510	560	625	645	660	810	850	890	910	950	970					
16	385	405	430	470	490	510	560	625	645	660	810	850	890	910	950	970				
17	385	405	430	470	490	510	560	590	625	645	660	810	850	890	910	950	970			
18	385	405	430	470	490	510	560	590	625	645	660	810	850	870	890	910	950	970		
19	385	405	430	470	490	510	560	590	625	645	660	810	850	870	890	910	950	970	1020	
No. of images	Selected wavelengths (nm)																			
1	RGB																			
2	RGB	970																		
3	RGB	405	1020																	
4	RGB	405	510	1020																
5	RGB	405	510	950	1020															
6	RGB	405	430	510	950	1020														
7	RGB	405	430	510	660	950	1020													
8	RGB	405	430	470	510	660	950	1020												
9	RGB	430	470	510	590	660	950	970	1020											
10	RGB	385	430	470	590	660	810	950	970	1020										
11	RGB	385	405	430	470	590	660	870	950	970	1020									
12	RGB	385	405	430	470	560	590	660	870	950	970	1020								
13	RGB	385	405	430	470	560	590	660	810	870	950	970	1020							
14	RGB	385	405	430	470	560	590	625	810	850	870	950	970	1020						
15	RGB	385	405	430	470	490	560	590	625	810	850	870	950	970	1020					
16	RGB	385	405	430	470	490	560	625	660	810	850	870	910	950	970	1020				
17	RGB	385	405	430	470	490	560	590	625	660	810	850	870	910	950	970	1020			
18	RGB	385	405	430	470	490	510	560	590	625	810	850	870	890	910	950	970	1020		
19	RGB	385	405	430	470	490	510	560	590	625	660	810	850	870	890	910	950	970	1020	
20	RGB	385	405	430	470	490	510	560	590	625	645	660	810	850	870	890	910	950	970	1020

## Data Availability

The data used to support the findings of this study can be made available by the corresponding author upon request.

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
