# Peer review of "Multispectral Food Classification and Caloric Estimation Using Convolutional Neural Networks"

_foods, 2023, doi:10.3390/foods12173212_

Round 1
Reviewer 1 Report
This paper integrates images with ultra violet, visible, and near infrared multispectral data, and establishes a method for food classification and calibration estimation using deep learning algorithms. This paper has innovation and research value.
One of the main research contents of this article is data fusion, that is, the fusion of RGB images and spectra. Therefore, it is recommended to highlight the keyword of data fusion in the title, abstract, and content of the paper.
Moderate editing of English language required.
Author Response
Comments and Suggestions for Authors
This paper integrates images with ultra violet, visible, and near infrared multispectral data, and establishes a method for food classification and calibration estimation using deep learning algorithms. This paper has innovation and research value.
One of the main research contents of this article is data fusion, that is, the fusion of RGB images and spectra. Therefore, it is recommended to highlight the keyword of data fusion in the title, abstract, and content of the paper.
- Reply: The author agreed with the reviewer’s opinion. And hence, “data fusion” was included in the list of keywords (marked in blue) and the sentence (line 3-5) was changes as “A method of fusion the RGB image and the images captured at ultraviolet, visible, and near-infrared regions at center wavelengths of 385, 405, 430, 470, 490, 510, 560, 625, 645, 660, 810, 850, 870, 890, 910, 950, 970, and 1020nm was adopted to improve the accuracy.” (line 4-6, marked in blue)
Reviewer 2 Report
The authors used multispectral images captured at ultraviolet, visible, and near-infrared regions with specific center wavelengths. They employed a convolutional neural network (CNN) to automatically classify food types and estimate their caloric content. The CNN was trained on a dataset containing 10,909 images from 101 food types. The objective functions, including classification accuracy and mean absolute percentage error (MAPE), were investigated based on the number of wavelengths used.
Question 1. Why were these 20 specific wavelengths (385, 405, 430, 470, 490, 510, 560, 590, 625, 645, 660, 810, 850, 870, 890, 910, 950, 970, 1020 nm, and white) chosen for acquiring multispectral images to differentiate food items and quantify food calories in this study? What is the rationale behind selecting these particular wavelengths? Could other wavelengths have been used instead?
Question 2. The structure of the paper is somewhat disorganized. For instance, all the data analysis methods are written within the same section.
Question 3. Is the food classification and calorie calculation performed simultaneously using a single model in this study? Please provide a detailed description of each layer structure and the parameters used in the model.
Question 4. This study only utilizes images from different bands (NIR, UV, or RGB images) as inputs to the model. Based on the discriminative accuracy, it is assumed that certain bands of images are more suitable for food classification and calorie calculation. However, is there any analysis conducted on the basis of food characteristics, nutritional substances, etc., to select the specific bands for this purpose?
Question 5. In the manuscript, there are multiple similar figures. It is recommended to consolidate these similar figures together for ease of comparison.
Question 6. The images captured in different spectral bands reflect various information about the food. In the manuscript, a substantial section is devoted to whether RGB images were used for classification and quantification. The significance of such exploration lies in understanding the potential contribution of RGB images in these tasks. However, it is indeed more crucial to identify and select specific spectral bands that carry more relevant information for food classification and calorie calculation. This approach would provide more valuable insights into the relationship between spectral bands and the respective tasks.
Minor editing of English language required
Author Response
Comments and Suggestions for Authors
The authors used multispectral images captured at ultraviolet, visible, and near-infrared regions with specific center wavelengths. They employed a convolutional neural network (CNN) to automatically classify food types and estimate their caloric content. The CNN was trained on a dataset containing 10,909 images from 101 food types. The objective functions, including classification accuracy and mean absolute percentage error (MAPE), were investigated based on the number of wavelengths used.
Question 1. Why were these 20 specific wavelengths (385, 405, 430, 470, 490, 510, 560, 590, 625, 645, 660, 810, 850, 870, 890, 910, 950, 970, 1020 nm, and white) chosen for acquiring multispectral images to differentiate food items and quantify food calories in this study? What is the rationale behind selecting these particular wavelengths? Could other wavelengths have been used instead?
- Reply: The selection of these wavelengths was made based on reference [24], “Raju, V. B; Sazonov, E. Detection of oil-containing dressing on salad leaves using multispectral imaging. IEEE Access, 2020, 8, 86196-86206.” In the original study [24], a total of 10 wavelengths (minimum of 380 nm and maximum of 980 nm) were used to detect oil on leaf surfaces. In this study, since the analysis of various foods should be carried out, the adjacent wavelength interval was further reduced, and 1020 nm was added to infrared band. As a result, 20 types of wavelengths, which are twice the number of wavelengths used in the previous study [14], were used in this study. As a follow-up to this study, experiments on a total of 29 wavelengths (250nm~1650nm) are in progress, and the results will be presented soon.
Question 2. The structure of the paper is somewhat disorganized. For instance, all the data analysis methods are written within the same section.
- Reply: The author thought that preliminary feasibility test was also part of analysis. Accordingly, section 3 (Preliminary feasibility tests for UV/NIR images) was moved to section 4 (Food analysis) as subsection.
Question 3. Is the food classification and calorie calculation performed simultaneously using a single model in this study? Please provide a detailed description of each layer structure and the parameters used in the model.
- Reply: Yes, the architectures of CNNs for food classification and caloric estimation were same each other, except for output layer (sigmoid and linear function for food classification and caloric estimation, respectively.) According to the reviewer’s comment, the author changed the figures of the CNN architecture (Figure 8) in which a detailed description (the kernel size of each layer and the size of the intermediate image, etc) is explained. (caption was also changed, marked in blue)
Question 4. This study only utilizes images from different bands (NIR, UV, or RGB images) as inputs to the model. Based on the discriminative accuracy, it is assumed that certain bands of images are more suitable for food classification and calorie calculation. However, is there any analysis conducted on the basis of food characteristics, nutritional substances, etc., to select the specific bands for this purpose?
- Reply: There were two possible methods in determining the set of wavelengths: early-determination and late-determination. Among them, early determination is a method of performing food classification and calorie estimation by pre-selecting specific wavelengths according to food characteristics, nutritional content, etc. as mentioned by the reviewer. A method of early determination has been applied to many food analysis techniques to date, as mentioned in the introduction of the paper (lines 91-93). This method requires prior knowledge of the optical properties (e.g., absorption/reflection spectra) of each of the ingredients of the food. To the best of our knowledge, such an approach is primarily applied to the analysis of single ingredients (e.g., water, sugar, soluble proteins, fats, etc.) rather than mixed ones.
Late determination refers to a method of selecting wavelengths without any prior optical characteristics of the underlying foods, solely from the perspective of minimizing or maximizing a particular metric. (e.g. cross-entropy or mean squared error) This method was adopted in this study. The reason for this is that 1) this study is not a quantitative analysis of a specific nutritional substance of food, but an analysis of a variety of foods, and 2) Since the calories (to be measured in this paper) are not determined by a specific component, but by a variety of components, it is difficult to limit the light source for caloric estimation to a specific band. Considering this aspect, the author used the late determination technique in which a piecewise selection method was adopted.
Question 5. In the manuscript, there are multiple similar figures. It is recommended to consolidate these similar figures together for ease of comparison.
- Reply: While Figures 9 and 10 in the paper look similar, they represent the food recognition rate and the margin of error in calorie estimation (MAPE), respectively. This indicates that the metrics of the presented results are different each other. Therefore, the author separated Figure 9 and Figure 10 rather than merging them. Both figures 12 and 13 look similar but represent the results for the case of food classification and caloric estimation. Since they are already presented back-to-back, it doesn't seem to make much sense to combine them into a single figure and separate them into top and bottom figures.
Question 6. The images captured in different spectral bands reflect various information about the food. In the manuscript, a substantial section is devoted to whether RGB images were used for classification and quantification. The significance of such exploration lies in understanding the potential contribution of RGB images in these tasks. However, it is indeed more crucial to identify and select specific spectral bands that carry more relevant information for food classification and calorie calculation. This approach would provide more valuable insights into the relationship between spectral bands and the respective tasks.
- Reply: The author believes this question is similar to question #4. The main interest of this paper is to improve the accuracy of food classification and calorie estimation by adding images acquired from different wavelengths of light sources to the RGB images that were popularly employed in the previous studies. To make this clear, the authors added the following sentence to abstract; “A method of fusion the RGB image and the images captured at ultraviolet, visible, and near-infrared regions at center wavelengths of 385, 405, 430, 470, 490, 510, 560, 625, 645, 660, 810, 850, 870, 890, 910, 950, 970, and 1020nm was adopted to improve the accuracy.” (line 4-6, marked in blue). In this paper, author is not interested in validating the usefulness of RGB images for food classification and caloric estimation, but rather in the complementary use of images of different wavelengths.
According to the reviewer’s comments that identification of a specific spectral bands is more crucial, the author conducted the experiments which involved a slight modification of feature selection method so that spectral band was constructed rather than individual wavelengths. Briefly, the experimental results showed that spectral band selection method revealed lower accuracy compared with individual wavelength selection, as follows:
|
Method |
Wavelength selection (RGB+1 Image(890nm)) |
Wavelength selection (Best, RGB+14 Images) |
Spectral band selection |
|
Food classification (%) |
95.5 |
99.5 |
94.5 |
|
Caloric estimation (MAPE) |
19.8 |
10.5 |
22.2 |
Moreover, light sources that emit a specific band of light either use LEDs of multiple wavelengths within the band or use halogen lamps and optical filters, which is a bulky and costly method compared to using RGB images and one NIR image (in this case only two light sources are needed: a white LED and an LED of the corresponding wavelength).
Obviously, identifying suitable spectral bands for food classification and caloric estimation is critical for light-based food analysis. This paper, however, is aimed at improving the accuracy of food classification and caloric estimation. (rather than finding the relevant spectral bands for food classification and caloric estimation) Although the explicit relationship between spectral bands and the respective tasks was not sufficiently addressed in the paper, the author believes that Tables 4 and 5 and the contents of Section 4.3 provide information on the most relevant wavelengths (albeit in spectral bands) for food classification and calorie estimation, respectively.
Reviewer 3 Report
The manuscript "Multispectral food classification and caloric estimation using convolutional neural networks" is interesting, since it uses multispectral imagens and convolutional neural netoworks to improve the quality of classification of foods and predictions of calories for commum food in Korea. As was showed by the authors both objectives were achieved.
My only question about the methodology is whether using information from reference values would be suitable as labels for the actual foods analyzed. But, I admit that the idea of looking for reference values for foods was a good strategy to use the available data. In addition, in a next study, it will be possible to test the trained model in other samples of food with accurately analyzed calorie values to evaluate the robustness of the model.
Follow bellow some suggestions for the authors concerning the text:
Abstract: the objective of the manuscript is not clear in the abstract;
Line 12 – 14: results need to be more clear written.
Linha 15: include a conclusion related to your objective in the abstract.
Line 16: exclud the keyword “obesity screenig”, since the main objective is not obesity screening
Line 117: change “aboive” to “above”.
Line 143: Change “the” to “The”.
Line 171: a do tis missing in this line.
Line 295: “latte” and “latte with sugar” are not presented in the table 1.
The english is good, but a minor review is needed to exclude minor text errors.
Author Response
Comments and Suggestions for Authors
The manuscript "Multispectral food classification and caloric estimation using convolutional neural networks" is interesting, since it uses multispectral imagens and convolutional neural networks to improve the quality of classification of foods and predictions of calories for common food in Korea. As was showed by the authors both objectives were achieved.
My only question about the methodology is whether using information from reference values would be suitable as labels for the actual foods analyzed. But I admit that the idea of looking for reference values for foods was a good strategy to use the available data. In addition, in a next study, it will be possible to test the trained model in other samples of food with accurately analyzed calorie values to evaluate the robustness of the model.
- Reply: The author agrees that chemical analysis methods could provide very accurate reference values. However, the food used in this study is a mixture of several ingredients, with the exception of a few. These foods may have different calories depending on the sample area. Reference calories for foods with a mix of ingredients can be obtained by averaging the calories in samples taken at different locations. However, the average calories obtained may fluctuate depending on the number of samples and the location of the sample. The calories in foods provided by Nutrition facts or the Korean Food Safety Administration are known to be measured considering the several factors, including those mentioned above. Therefore, rather than measuring the reference calorie values for each food by the author himself (which may be less accurate), the author decided to use the calorie values that have already been published.
Follow bellow some suggestions for the authors concerning the text:
Abstract: the objective of the manuscript is not clear in the abstract;
- Reply: One of the main purposes of this paper is to improve the accuracy by using images captured under various light sources in the ultraviolet, visible, and infrared bands compared to conventional food classification/calorie measurement methods using RGB images. Accordingly, the author clearly mentioned the objective of the paper in abstract as follows: “A method of fusion the RGB image and the images captured at ultraviolet, visible, and near-infrared regions at center wavelengths of 385, 405, 430, 470, 490, 510, 560, 625, 645, 660, 810, 850, 870, 890, 910, 950, 970, and 1020nm was adopted to improve the accuracy.” (line 4-6, marked in blue)
Line 12 – 14: results need to be more clearly written.
- Reply: The author changed as “The highest accuracy for food type classification was 99.81% when using 19 images and the lowest MAPE for caloric content was 10.56 when using 14 images.” (line 14-16, marked in blue)
Linha 15: include a conclusion related to your objective in the abstract.
- Reply: A new sentence was included in abstract as follows: “Such results proved that the use of the images captured at various wavelengths in the UV and NIR bands was very helpful to improve the accuracy of food classification and caloric estimation.” (line 16-17, marked in blue)
Line 16: exclude the keyword “obesity screening”, since the main objective is not obesity screening
- Reply: The author removed the keyword “obesity screening”
Line 117: change “aboive” to “above”.
- Reply: The author changed it accordingly. (marked in blue)
Line 143: Change “the” to “The”
- Reply: The author changed it accordingly. (marked in blue)
Line 171: a do tis missing in this line.
- Reply: The author changed it accordingly. (marked in blue)
Line 295: “latte” and “latte with sugar” are not presented in the table 1.
- Reply: As the authors checked, both "caffelatte" and "latte" were used. It has been unified as "caffelatte" to prevent confusion among readers. (marked in blue)
Round 2
Reviewer 2 Report
The authors have responded in detail accroding to the comments requested and revised and improved the manuscript accordingly.